# MicroRNAs Regulate Cell Cycle and Cell Death Pathways in Glioblastoma

**DOI:** 10.3390/ijms222413550

**Published:** 2021-12-17

**Authors:** Isra Saif Eldin Eisa Sati, Ishwar Parhar

**Affiliations:** Brain Research Institute Monash Sunway [BRIMS], Jeffrey Cheah School of Medicine and Health Sciences, Monash University Malaysia, Bandar Sunway 47500, Malaysia; isra.sati@monash.edu

**Keywords:** RB, P53, PI3K, EGFR, MGMT, TMZ, autophagy, apoptosis, exosomal miRNAs

## Abstract

Glioblastoma (GBM), a grade IV brain tumor, is known for its heterogenicity and its resistance to the current treatment regimen. Over the last few decades, a significant amount of new molecular and genetic findings has been reported regarding factors contributing to GBM’s development into a lethal phenotype and its overall poor prognosis. MicroRNA (miRNAs) are small non-coding sequences of RNA that regulate and influence the expression of multiple genes. Many research findings have highlighted the importance of miRNAs in facilitating and controlling normal biological functions, including cell differentiation, proliferation, and apoptosis. Furthermore, miRNAs’ ability to initiate and promote cancer development, directly or indirectly, has been shown in many types of cancer. There is a clear association between alteration in miRNAs expression in GBM’s ability to escape apoptosis, proliferation, and resistance to treatment. Further, miRNAs regulate the already altered pathways in GBM, including P53, RB, and PI3K-AKT pathways. Furthermore, miRNAs also contribute to autophagy at multiple stages. In this review, we summarize the functions of miRNAs in GBM pathways linked to dysregulation of cell cycle control, apoptosis and resistance to treatment, and the possible use of miRNAs in clinical settings as treatment and prediction biomarkers.

## 1. Introduction

Glioblastoma (GBM) is a grade IV tumor, which develops from the brain’s supporting glia cells and represents around 70–75% of total glioma cases [1]. GBM’s incident rate is estimated to be 3.2 cases per 100,000 and its median survival is approximately 15 months, which accounts for a survival percentage of 5% for a maximal period of 5 years [2]. GBM is characterized by multiple genetic alterations, cell proliferation, necrosis, and microvascular proliferation [3]. These hallmarks of GBM lead to its lethal phenotype and tumor resistance to treatment, which is GBM’s main treatment challenge [3]. The majority of GBMs are primary or de novo GBM (accounts for over 90% of all GBM). At the same time, secondary GBM is less common and progresses from pre-existing anaplastic astrocytoma or low-grade diffuse astrocytoma [4,5,6]. Both primary and secondary GBMs display alteration in tumor Protein 53 (P53) pathway (87%), receptor tyrosine kinase pathway (RTK) (88%), and retinoblastoma pathway (78%) [7]. Dysregulation in these pathways promotes cell proliferation, cell cycle checkpoints escape, and cell survival. In GBM, guardian molecules of these altered pathways are mutated or ineffective due to the high mutation nature of the tumor’ [1,8]. Gene profiling of GBM reveals a number of genes to be highly dysregulated, including P53, phosphatase and tensin homolog (PTEN), retinoblastoma RB, survivine (BIRC5), and O^6^-methylguanine-DNA methyltransferase (MGMT) [9]. Amplification of the epidermal growth factor receptor (EGFR) also impacts pathways linked to cell death [10]. Furthermore, over-activity of pathways like DNA repair, P13K-Akt pathway, or distribution of proteins related to mitochondrial apoptosis aid the growth of an unwanted population of cells, which eventually leads to treatment failure [7]. The involvement of multiple pathways in GBM tumorigenesis and progression highlights its complexity and heterogenicity.

In recent years, there has been an emerging interest in microRNAs (miRNAs) as curial factors for multiple roles as oncogenes or tumor suppressor genes [11]. The ability of miRNAs to function at a transcriptional level in multiple cancers makes them potential candidates for studies related to the progression and treatment of GBM [12]. This review focuses on the possible effect of miRNAs on pathways that are altered in GBM, leading to its lethal phenotype. Furthermore, this review introduces the potential use of miRNAs as biomarkers and predictors of effectiveness and response to treatments, aiding the current knowledge and development of GBM standards of care.

## 2. MiRNAs and Their Role in Cancer Development

### 2.1. MiRNA: Structure and Biogenesis

MiRNAs are natural anti-sense interactors, transcribed from primary transcripts (Pri-miRNAs) and are highly regulated by RNA polymerase II and promoter elements [13]. The primary transcripts of MiRNAs can contain a sequence for multiple or single miRNAs [14]. Mature miRNAs are usually less than 25 nucleotides in length; they are pri-miRNA processed through RNase III in the nucleus and exported to the cytoplasm [15]. Mature miRNAs form hairpin loop structures, which allows them to interact with mRNA, partially binding to its complementary sequence at the mRNA leading to inhibition of targeted genes, translation process or deregulation of related mRNA [12]. The modulatory potential of miRNA is exceptionally high; over 25% of genes expressed in humans are regulated by miRNA [16]. In total, over 1400 miRNA have been discovered, and each may target multiple mRNAs. It is predicted that miRNA can control the activity of over 60% of mRNAs [17,18]. MiRNAs are grouped in families with similar and sometimes the same basic sequence, which explains multiple miRNAs targeting the same target [19]. Nevertheless, it is unlikely a single miRNA to change the outcome of mRNA via a single target, but rather a single miRNA could have multiple targets of related mRNA to maximize its functional capacity [19]. MiRNAs are regulated by their transcription units; any dysregulation of these units can lead to deregulation of their gene expression levels, a single unit can activate or deactivate many of its miRNA target genes [19]. This circuit negatively regulates itself. MiRNAs control specific genes either by increasing or decreasing their expression and the products, miRNAs can control the gene expression level of other miRNAs. The shortcomings of these circuits impact this tightly regulated system, affecting major signaling pathways regulated by these miRNAs [19,20]. Collectively, these facts make miRNA an exciting factor to look at its possible function in healthy biological status and in malignancies.

### 2.2. MiRNA: Link to Cancer Development

MiRNA–mRNA interactions regulate the expression of specific proteins related to normal biological functions like cell survival, developmental timing, cell cycle regulation, proliferation, and apoptosis [21,22]. MiRNAs also regulate hematopoietic and adipocyte differentiation and insulin secretion [23,24,25]. Many of these vital functions, when altered, lead to the initiation and development of tumors [26,27]. Furthermore, over 50% of miRNA genes are within the exons of protein-coding genes located in fragile regions, cancer-specific translocation breakpoints, which are highly involved in cancer progression and development [13,28,29]. Either genetic loss or epigenetic silencing can alter the expression level of miRNAs [30]. MiRNAs can act in a cascade of events leading to oncogenesis by acting as tumor suppressors or as oncogenes by alternating protein-coding oncogenes or tumor suppressor genes; this highlights their critical role as mediators at transcriptional and post-transcriptional levels [31]. The amplification of regions encoding for oncogenic miRNAs leads to an up-regulation of oncogenic miRNAs and down-regulation or silencing of tumor suppression genes. Conversely, the deletion or mutations of a region containing tumor-suppressing miRNAs leads to its downregulation but upregulation of oncogenic genes.

MiRNAs function in context-dependent manner, a single miRNA species is not specific to a particular function but depends on, in which tissue it is expressed and its impact on pathways altered in that tissue [17,19]. An example of this is miR-29, it functions as an oncogene in breast cancer but as a tumor suppressor gene in lung-related tumors [31,32]. Some miRNAs are unique to GBM, while others are also found to be involved in other tumors. MiR-26a is up-regulated in glioma but downregulated in hepatocellular carcinoma [33,34,35]. As reported in breast cancer studies, the difference in miRNAs expression depends on the molecular subtype of the tumor it is expressed in, which also aids in the classification of tumor malignancies [14].

### 2.3. MiRNA: Link to GBM

The dysregulation of miRNAs can negatively influence the pattern of cell transformation affecting the development of the normal brain [36]. MiRNAs influence cell identity in the brain and are involved in stem cell differentiation [37,38]. The pattern of miRNAs expression during brain development is different between normal tissues and tumors; this reflects their possible role in developing and progressing CNS-originated tumors [39,40]. Furthermore, the different miRNA expression pattern also reflects on the developmental lineage of GBM [41]. As such, miRNAs can serve as a switch in the mammalian differential pathway, and protein production and altered expression can contribute to tumor development, including GBM [13]. Many miRNAs are involved in cancer and its development through cancer-related pathways, including cell cycle control, apoptosis, and DNA damage response [12]. So, either the loss or overexpression of specific miRNAs can positively contribute to survival pathways and disturbance of the cell cycle [12,31]. In GBM, miRNAs act on genetic pathways which are already altered, including the P53 pathway, RB pathway and PI3K-Akt pathway. This excessive alteration aids GBM tumorigenesis, which includes increased proliferation rate, decrease in apoptosis, de-diffraction, angiogenesis, and metastasis. The alteration of pathways related to apoptosis in GBM are further elaborated, and miRNAs involved are highlighted (see below).

## 3. MiRNA Actions on GBM Altered Pathways

MiRNA’s link to the progression of GBM was first identified to miR-21. Chan and co-workers showed the expression of miR-21 highly elevated in GBM cells and adult malignant brain tissues, suggesting its possible role as a micro-oncogene and anti-apoptotic factor [42]. Furthermore, the discovery of miR-21 in GBM suggested its potential role in GBM malignant phenotype [42]. It is important to note that miR-21 is not unique to GBM; its altered expression level is seen in multiple tumors types, including breast, lung, and colon [43]. MiR-21’s link to mechanistic function in GBM was first reported by Papagiannakopoulos using pathway analysis. Their study showed a direct link between the ability of miR-21 in limiting apoptosis by altering the activity of P53 and TGF-beta [44]. Furthermore, the same study also reported that restoration of MiR-21 expression to baseline levels led to inhibition of cell growth, proliferation, and survival [44].

## 4. MiRNA in GBM Cell Cycle Control

### 4.1. Deregulation of P53 Pathway

P53 is a well-known tumor suppressor and transcription factor. Its biological functions include cell cycle arrest, apoptosis, genome stability, cellular metabolism, and angiogenesis inhibition [45,46]. P53 pathway, which includes mouse double minute 2 homolog (MDM2), ADP-ribosylation factor (ARF), Cyclin-Dependent Kinase Inhibitor 2A (CDKN2a), and P53 as a key regulator. This pathway is deregulated in over 80% of GBM patients and in around 94% of GBM derived cells [47,48]. The alteration of this pathway includes deletion of *CDKN2A/ARK* and amplification of both MDM2 and MDM4 [7,49,50]. In percentages, 60% of *ARF* is found to be mutated in GBM, while 22% and 10% for *TP53* and *MDM2*, respectively [49,50]. All of which implicates cell cycle progression and uncontrolled growth of an unwanted population of cells with damaged DNA [46]. P53 is tightly regulated by PTEN by a negative feedback loop; even with a non-mutated version of P53, overexpression of PTEN can lead to loss of P53 function [46]. In GBM, the mutation of both P53 and PTEN is required to drive GBM progression and the ability to escape the process of cell death [46].

Since cancer is a multifactorial disease, the high expression of specific miRNAs impacts these pathways and contributes to its development. For example, both miR-21 and miR-26 are overexpressed in GBM can act on mRNA of many genes related to P53 [42]. The low expression of miRNAs can result in the inhibition of cell cycle arrest and cell death. For example, restoration of low levels of miR-126 to normal levels in GBM significantly decreased MDM2 expression and increased P53 protein expression level [51]. This restoration of miR-126 expression remarkably inhibited GBM cell proliferation, migration, and invasion and promoted apoptosis [51]. Other miRNAs that positively regulate the expression of P53 include miR-25 and miR-32, the forced expression of both miRNAs directly targets both MDM2 and TSC1 [52]. This negative regulation of both P53 pathway and mTOR pathway leads to P53 accumulation, cellular growth inhibition, and increased cell death [52]. It is worth noting that both miR-25 and miR-32 expression are auto-regulated by transcription factors E2F, MYC, and P53 [52].

Moreover, miR-34 is a tumor suppressor miRNA family; the expression level of miR-34a is notably decreased in GBM [53,54,55]. MiR-34 is reported to be transactivated and regulated by P53, and its level of expression is higher in GBM cells with wild type P53 than cells with mutated P53 [53]. It is also reported that the expression of miR-34a negatively correlates with the expression of c-Met in GBM cells [53]. Furthermore, miR-34a targets the Notch pathway in GBM cells and triggers the expression of both c-Met and Notch, inhibiting in vivo growth of GBM tumor and inducing cell death [53]. Notch pathway functions differently depending on the tumor type it activates. Notch pathway can suppress or increase the activity of P53, indicating its possible interconnection with P53 pathway [56]. This indicates the possibility of miRNAs acting as an extra factor to force specific pathways to be favored over others.

### 4.2. Deregulation of Retinoblastoma (RB) Pathway

RB is a phosphoprotein that functions to suppress the cell cycle at its early phase (G1-S phase) [57]. Rb is regulated by cyclin-dependent kinases (CDKs) with their cyclin partners, Cdk4 and Cdk6, and Cyclin E- CDK2. Rb is a self-regulated E2F family of transcription factors; suppressing its function leads to cell cycle arrest [57]. MiRNAs also regulate RB pathway. The expression level of both miR-124 and miR-137 is downregulated in GBM. Restoring their expression level leads to increased cell cycle arrest at G0/G1 phase by CDK6 expression [41,58]. Both miR-34a and miR-128, act on E2Fa and CDK6, respectively, as a tumor suppressor miRNAs [54,59]. Overexpression of miR-34a in GBM cells results in cell cycle arrest at G1 phase by negatively regulating cyclin E2, CDK4, and CDK6 and downregulation of the transcription factor E2F3 [54]. Furthermore, the expression of miR-128 inhibits cell proliferation and is negatively correlated with the expression level of E2F3a [59].

## 5. MiRNA Action on PI3K-Akt Pathway

Another extensively altered pathway in GBM is the PI3K-AKT-PTEN pathway. In this pathway, PTEN acts as a negative regulator to the PI3K–AKT interaction, causing G1 cell cycle arrest and apoptosis [9,60]. In GBM, this pathway is overly activated due to the loss of tumor suppresser PTEN, which is mutated in around 25% of primary GBM [4,5]. PTEN loss of function does implicitly the pathogenesis of GBM due to uncontrolled activation of PI3K-AKT pathway resulting in cell survival and proliferation [3]. PI3K pathway is influenced at multiple levels by a set of miRNAs, including miR-21, miR-7, and miR-126. MiR-21 can act as an oncogene in GBM by influencing and activating P13K/AKT, whereas silencing miR-21 inhibits this pathway and enhances autophagy activity [61]. Overexpression of miR-26 is associated with monoallelic loss of PTEN, promoting GBM tumorigenesis [34]. On the other hand, overexpression of both miRN-21 and miRN-26 can be used for GBM prognoses and as a diagnostic marker of high-grade glioma. Moreover, miR-126 is a tumor suppresser gene that acts on insulin receptor 1 (IR1), which is located upstream and one of the direct effectors of PI3K-AKT pathway [62]. Restoration of miR-126 expression level to baseline can lead to a decrease in P13K, and p-AKT level [51]. Similar outcomes are seen in an in vivo model when miR-126 along with LncRNA-XIST (long non-coding RNA-X inactive specific transcripts) have been associated with proliferation and GBM cell tumorigenesis in nude mice [63]. These studies do highlight the possibility of miRNAs interacting with other non-coding RNA sequences.

EGFR, a member of RTK family, upon activation, EGFR can trigger many pathways including of PI3K-AKT pathway [64]. Multiple variants of EGFR are reported in GBM, including *EGFRvI*, *EGFRvII*, *EGFRvIII*, *EGFRvIV,* and *EGFRvV*. Among all variants, *EGFRvIII* is the most common, which is oncogenic and functions by keeping the receptor in an active status without binding to its ligand. The overexpression of wild-type EGFR and EGFRvIII are responsible for the overly active downstream signaling, which leads to an increase in GBM proliferation and survival. In a GBM tumor, wild-type EGFR and *EGFRvIII* are *co*-expressed [65]; 97% of GBM cells express wild-type EGFR while 23% of GBM show a EGFRvIII mutation, which makes these two forms of EGFR the clinically most relevant [66]. Xu et al. recently reported on all miRNAs that act directly and indirectly on EGFR [67]. Among them, only one report linked the low expression of two miRNAs (miR-524-2P and miR-524-3P) to the overexpression of EGFR and EGFRvIII; other reports refer to EGFR in a general terms without specifying the variants [68]. Low expression of miR-7 is linked to elevated expression of EGFR [69]. Recent reports highlighted miR-7 ability to function independently of EGFR and EGFRvIII because the 3′-URT region of EGFR lacks miR-7 recognition site [70]. MiRNA expression profiles suggest different action of the same miRNAs is highly dependent on the miRNAs that are over-expressed in the GBM. [71]. MiR-7 also down-regulates (Insulin Receptor Substrate 2) IRS2, PI3K mediator, and AKT activity [69]. The expression of miR-200a-3p is downregulated in high-grade glioma [72] and is negatively correlated with G protein α inhibitory subunit 1 (Gαi1). Gαi1 is a mediator of P13K-Akt-mTor pathways by activating EGFR [73]. Low expression of miR-200a-3p in GBM leads to Gαi1 overexpression, leading to an increase in cell proliferation and escape from cell apoptosis by Akt activation [72,73]. Furthermore, the expression level of miR-126, located in the introns of EGFR like-domain, is low in GBM patients and the deletion of miR-34a in GBM correlates with amplified EGFR in vivo and in vitro [51,55].

## 6. MiRNA in GBM Cell Death Pathways

### 6.1. MiRNA in GBM Mitochondria-Mediated Apoptosis

Mitochondrial apoptosis is extensively studied in cancer, and its critical mechanistic parts are well defined and understood. Upon initiation by apoptotic stresses, like activation of death receptors or premobilization of mitochondria proteins, caspase proteins will accrue, inducing pro-caspase 9, caspase-3, and caspase-7 activation. The activity of caspases leads cells to immediate cell death due to the cleavage of multiple related substrates [74]. As this process is a point-of-no-return, mitochondrial apoptosis activity is tightly regulated by three sets of proteins: pro-apoptotic effector proteins, pro-apoptotic protein, and anti-apoptotic proteins. All these proteins are part of a larger family called BCL-2 (B-cell lymphoma 2) protein family [75]. Mitochondrial apoptosis is the most deregulated cell death in many types of cancers, and multiple miRNAs trigger and affect upstream regulator proteins of mitochondrial apoptosis. For instance, miR-128, a brain-specific miRNA, is down-regulated in GBM tissues [22]. Its overexpression decreases the level of cell proliferation and induces apoptosis by up-regulating BAX, downregulating BCL2; and increasing the level of cleaved caspase 3 [58,76]. The elevated expression of miR-181a acts as a suppressor to lower the expression level of BCL-2 gene and its protein product [77,78]. Low expression of BCL-2 triggers the activation of apoptosis, leading to an increased sensitivity to radiotherapy but has no impact on GBM response to chemotherapy [77]]. MiR-21 knockdown studies in vitro show increased caspases activity and increased apoptosis, which confirms the role of miR-21 as an anti-apoptosis factor, acting in multiple pathways [42]. The interaction of multiple miRNAs towards both BCL2 and BAX is clearly demonstrated in Figure 1.

### 6.2. Deregulation of Autophagy

Autophagy is a multiple lysosomal degradation processes responsible for the removal of unnecessary or dysfunctional cellular components. This process gets activated as a response to stress, leading to a breakdown of intracellular organelles into their building blocks (amino acids), and the amino acids are recycled to build proteins. In cancer, the process of autophagy protects by promoting cellular proliferation and differentiation. In addition, autophagy contributes to cancer progression by degrading major apoptotic mediators leading to reduction or inhibition in natural cell death. In GBM, many miRNAs are reported to regulate autophagy at multiple stages (summarized in Figure 2).

### 6.3. Deregulation of DNA Repair System

Current standard treatment for GBM follows a multi-modality approach, and patients typically undergo multiple treatment phases, maximal surgical resection followed by concurrent radiotherapy and Temozolomide (TMZ) treatment [7,79]. TMZ, a DNA alkylating agent, works by inducing cell-cycle arrest leading to cell death. Even though this standard treatment improves the patient’s survival rate, it is not curative, and 90% of patients eventually develop resistance towards treatments [16]. The primary regulators of DNA repair system are O(6)-methylguanine-DNA methyltransferase (MGMT), mismatch repair (MMR), and base extension repair (BER) [80]. DNA repair proteins naturally work to remove incorrect bases added to the DNA pairs by external environmental factors, such as oxidizing agents, ionizing radiation, or alkylating agents, allowing direct repair of the created lesions [81,82]. The level of DNA repair molecules varies across different types of tumors; an activated DNA repair system suppresses normal cell death process [83]. MGMT is rarely mutated or deleted; its elevated expression in GBM is mainly linked and controlled by the methylation of its promoter [84]. Hyper-methylation of the MGMT promoter is found in 45% of GBM patients; they show a better response to TMZ [85]. GBM patients with low expression of MiR-181b and MiR-181c respond better to TMZ treatment by acting directly on MGMT promoter [86]. Furthermore, MGMT indirectly affects the control of miR-17 through blockage of STAT3 activity; this works as a negative regulator of MGMT as its expression level is directly linked to STAT3 activation [87]. Further, MGMT is a direct target of miR-370-3p, a high expression of miR-370-3p can alter the expression of MGMT, restoring GBM cell sensitivity towards TMZ [88]. MiRNAs have an indirect impact on resistance, for example, miR-21 act on mitochondria apoptotic proteins that protect GBM cells from the potent TMZ damage by decreasing caspase-3 activity and the Bax/Bcl2 ratio [89]. The target genes of all miRNAs described above are summarized in Table 1.

## 7. MiRNA in GBM Inflammation

Inflammation is another process that is mediated by miRNAs. During this process, many inflammatory mediators within the tumor cells as well as in the tumor microenvironment are off-balance, and this can yield an attractive environment for tumor initiation and growth [91]. Even though the entire mechanism of inflammation in GBM is uncertain, it is clear that the outcome of inflammation does aid GBM survival, proliferation, and maintenance [92]. The immune system in GBM is an interactive scene when multiple types of cells are involved, such as tumor cells, immune cells, and antigen-presenting cells. The unique genetic and cellular environmental status of GBM support its ability to escape immune suppression by the over-activated immune checkpoints. GBM is mediated by many immune checkpoints that favor GBM maintenance. Furthermore, GBM is considered as a ‘cold tumor’ when the level of activated T cells is low, which prevents the antitumor immune response [93]. Many reports highlight the interaction between miRNAs, inflammatory mediators, and cellular environmental checkpoints that affect the expression of miRNAs. Immunological miRNAs act as part of the immune system antitumor process [94]. The immunosuppressive environment of GBM, which is hypoxic and rich with reactive oxygen specious, can highly influence the biogenesis and activity of miRNAs [94]. At the same time, this environment does increase the expression of hypoxia-induced transcription factors such as HIF1α that act as a driving force to increase the expression of many hypoxic adaptation-related miRNAs [95,96]. Up to date, many immune checkpoints have been highlighted in GBM tumors. One well studied immune checkpoint is programmed cell death (PD-1), presented on immune cells surface and its ligand-programmed cell death ligand 1 (PD-L1), which is present on GBM tumor cells. The ability of GBM tumor cells to activate death receptors on the immune cells gives GBM the privilege to escape T cell killing [97,98]. The majority of miRNAs act as anti-inflammation tumor suppressers, by directly inhibiting the expression of PD-L1 on GBM cells, these miRNAs include miR-424, miR-138-5p, miR-34a, miR-200, and miR-513. The expression of PD-L1 is indirectly regulated through interactions with various miRNAs, negatively by PTEN, and positivity by STAT3. Furthermore, PD1 expression is negatively regulated within immune cells by miR-28 and miR-138 [12]. Another immune checkpoint is cytotoxic T-lymphocyte-associated protein 4 (CTLA4), when activated, can downregulate the immune response in GBM. CTLA4 is present on the surface of immune cells and becomes activated with its ligand, CD80 (the cluster of differentiation 80). Both CTLA4 and CD80 expression can be inhibited by miRNAs. CTLA4 is inhibited by miR-138 and miR-155 and CD80 expression is inhibited by miR-424 [12]. Moreover, GBM secretes many pro-inflammatory mediators into the tumor microenvironment, and these mediators can create a reinforcing loop that keeps the inflammatory status of GBM active. Recently reported, miR-93 expression negatively regulates the expression of many pro-inflammatory mediators like HIF1α, mitogen-activated protein kinsae2 (MAP3K2), interleukin 6 (IL-6), interleukin 8 (IL-8), and leukemia inhibitory factor (LIF) [99]. miR-93 can also indirectly act to regulate the expression level of cyclooxygenase 2 (C0X2) and C-X-C motif chemokine 5 (CXCL5) via HIF1α and MAP3K2 [99].

## 8. Free Circulating MiRNAs as GBM Diagnostic Tool

Exosomes are molecular cargo that carries a mix of many biomolecules, including DNA, proteins, lipids, mRNA, and non-coding RNA. Exosomes are released from normal and malignant cells into the extracellular space as nano-sized vesicles, which are unique with molecular characteristics of their cell of origin [100]. Since exosomes can cross the blood–brain barrier into the peripheral circulation with their molecular cargo, identification of these free-circulating exomes can be a non-invasive method to detect GBM’s genetic makeup and progression. MiR-128 and miR-342-3p from the peripheral blood have been associated with GBM diagnosis [101]. According to WHO 2016 classification, the molecular grouping of GBMs is based on mutations of isocitrate dehydrogenase (IDH) [1,6]. Recently, Seven miRNAs have been identified (miR-182-5p, miR-328-3p, miR-339-5p, miR-340-5p, miR-485-3p, miR-486-5p, and miR-543), and these seven miRNA show over 91% accuracy to detect GBM in wild type IDH, which overlaps with primary GBM [102]. Five miRNAs associated with lower grade and mutated IDH glioma overlap with secondary GBM, these miRNAs include miR-7d-3p, miR-106b-3p, miR-130b-5p, miR-185-5p, and miR-98-3p [102]. Mechanistically, these signature miRNAs deregulate IDH itself; however, more studies are needed to identify key regulators and their effects [102]. Furthermore, the high-expression levels of miR-182-5p and miR-486-5p are critically linked to GBM tumor growth, survival, and promoting GBM aggressiveness in vitro [102]. MiR-340-5p, miR-485-3p, and miR-543 are tumor suppresser genes; both were in extremely low abundance in GBM tumor compared to normal brain [102]. Lastly, the decreased expression of miR-339-5p modulates T cells’ response, leading to immune evasion of GBM [102]. Together with the WHO classification, free-circulating exosome miRNAs can be an accurate detection tool to predict the genetic background of each GBM patient. This can serve as a guide to personalize treatment plans to ensure greater benefits for every patient.

## 9. Possible Use of miRNAs to Predict GBM Treatment Efficiency

GBM has always been a challenge when it comes to its diagnosis and treatment. Even though glioma cases only account for approximately 1% of global cancer cases, it has a low survival rate and resistance to treatments [1,9,103]. Although the current treatment regimen of GBM improves the patient’s survival, over 90% of GBM cases do not respond to the second cycle of chemotherapy [16]. An in-depth understanding of the molecular mechanisms of miRNAs in GBM is essential. Genetic profiling has been used extensively to map GBM specific miRNAs as predictors for treatment efficacy and outcome. Besides their use as a diagnostic marker, miRNA can also be used as treatment outcome predictors. Recently identified, the Sonic hedgehog pathway is overly activated in TMZ resistance GBM cells, while the expression of hedgehog suppresser receptor, PTCH1, is decreased [104]. The low expression of PTCH1 is due to an increase in miRNA-9, which increases the expression level of drug efflux genes MDR1 and ABCG2 [104]. The high expression levels of miRNA-9 are liked to recurrent GBM tumors and not to the primary tumor. Furthermore, a set of miRNA can serve as predictors to TMZ sensitivity in GBM cells including, miR-21, miR-370-3p, miR-423-5p, miR-181, miR-195, miR-455-3p, and miR-10a [89,105]. These reports collectively highlight the future use of miRNA, along with already established biomarkers, for GBM diagnosis and as part of GBM immunotherapy.

## 10. Viability of Targeting miRNAs in GBM Clinical Settings

Apart from using miRNAs in GBM diagnosis, miRNAs show potential as a therapeutic target. Due to their small size and stability, miRNAs have an advantage over protein-coding genes in GBM treatment. MiRNAs based drugs are either mimics or inhibitors to target genes that dysregulate signaling pathways. The expression of oncogene miRNA can be inhibited by synthetic anti-sense such as single-stranded RNA-based oligonucleotides (antagomirs or anti-miRNA) or miRNA sponges [106]. The three main considerations for using miRNAs in therapies include: (1) Delivery method. Even though miRNAs are small in size and stable in serum, systemic delivery is not the best approach to achieve high efficiency. The convection-enhanced local delivery and the nanoparticle-based delivery are better approaches for the stability and the efficiency of the miRNA across the blood–brain barrier to the site of the tumor cells with least interaction with health cells [106]. (2) Specificity of the target site. GBM specific surface receptors serve as recognition targets to ensure the specificity of miRNA delivery to the correct location. (3) Specificity of the gene of interest [106]. MiRNA’s ability to target multiple genes is a limiting factor for its therapeutic use. The main obstacle for miRNA-based drugs is targeting the gene of interest to avoid unspecific targets [107]. Ten miRNA based drugs have entered clinical trials but none for GBM [107]. This highlights the fact that miRNA-based therapies cannot be ‘one size fits all as the outcome of the same drug is likely to differ due to the many targets for the same miRNA. This can trigger unwanted pathways and unpreventable consequences. More studies are needed to increase the specificity of miRNAs-based drugs to target the desired gene(s) of interest.

## 11. Conclusions

The ability of miRNAs to regulate and influence the expression of multiple genes highlights their importance in facilitating and controlling normal biological functions as well as the initiation and progression of cancer. In this review, we have highlighted the role of miRNAs as tumor suppresser genes or oncogenes in pathways linked to GBM progression, including P53, RB, and PI3K pathways. These pathways were precisely found to be linked to cell proliferation and resistance to treatment of GBM. The growing interest in non-coding transcripts, including miRNA, can provide a better understanding of cancer gene modulation at transcription and post transcription levels. This will increase and strengthen the pool of knowledge, which could help to develop miRNAs as treatment and prediction biomarkers for GBM.

## Figures and Tables

**Figure 1 ijms-22-13550-f001:**
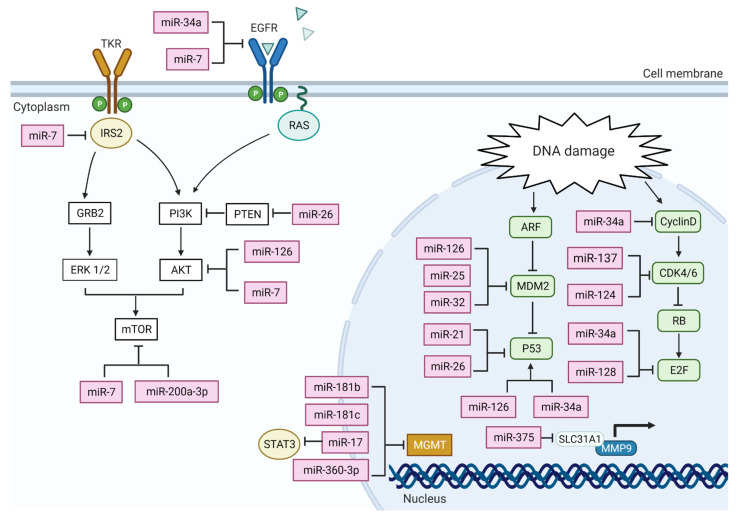
A diagram of miRNAs associated with major altered pathways reported in GBM development, including the P53 pathway, RB pathway, and P13K pathway. The function and the interaction of these miRNAs are based on their mechanistic studies following their baseline expression in GBM cells and tissue. An arrow (→) out of any miRNA represents a direct activation towards a targeted gene. For example, in the nucleus, it is reported that miR-126 targets P53, and an increase in miR-128 expression leads to an increase in the expression level of P53. Conversely, the symbol (⊢) represents an inhibition by miRNA towards a targeted gene. For example, an up-regulated expression of miR-34a leads to an inhibition in the expression of cyclin D gene. Figure created with BioRender.com.

**Figure 2 ijms-22-13550-f002:**
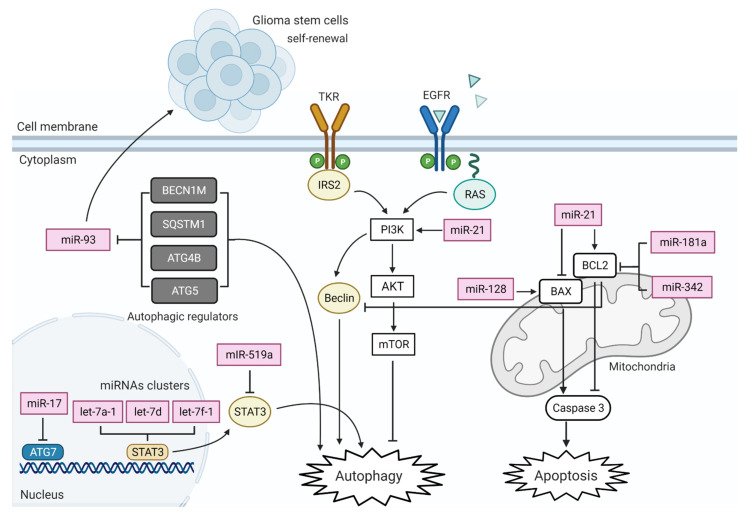
A diagram of miRNAs associated with major death pathways in GBM development, mitochondrial apoptosis, and autophagy. The function and the interaction of these miRNAs are based on mechanistic studies following their baseline expression in GBM cells. An arrow (→) out of any miRNA represents a direct activation of a targeted gene. For example, silencing miR-21 increases autophagy activity and autophagosome formation in GBM cells via direct inhibition of PI3K-Akt. Conversely, the symbol (⊢) represents an inhibition by miRNA of a targeted gene. For example, an up-regulated expression of miR-17 leads to inhibition of ATG gene expression. Figure created with BioRender.com.

**Table 1 ijms-22-13550-t001:** The targets of multiple miRNAs on altered pathways in GBM.

MiRNA	Pathway Affected	Target Gene/s	Expression	Reference
**mir-137**	CDKs-RB-E2F	CDK6	downregulated	[41,58]
**mir-124**	CDKs-RB-E2F	CDK6	downregulated	[41,58]
**mir-34a**	CDKs-RB-E2F	CyclinD1, cyclinE2, CDK4/6	downregulated	[54]
**mir-128**	CDKs-RB-E2F	E2F3a	downregulated	[59]
**mir-375**	SLC31A1-MMP9	SLC31A1	downregulated	[90]
**mir-21**	ARF-MDM2-P53	P53	upregulated	[42]
**mir-26**	ARF-MDM2-P53	P53	up-regulated	[34]
**mir-126**	ARF-MDM2-P53	MDM2-P53	downregulated	[51]
**mir-126**	PTEN-PI3K-Akt	P13K-Akt	downregulated	[51]
**mir-25**	ARF-MDM2-P53	MDM2-TSC1	downregulated	[52]
**mir-32**	ARF-MDM2-P53	MDM2-TSC1	downregulated	[52]
**mir-34a**	ARF-MDM2-P53	P53	downregulated	[53,54,55]
**mir-34a**	Notch pathway	c Met, Notch	downregulated	[53,54,55]
**mir-21**	Mitochondrial apoptosis	Caspase 3, Bax/BCl2	upregulated	[89]
**mir-181b**	DNA repair	MGMT	downregulated	[86]
**mir-181c**	DNA repair	MGMT	downregulated	[86]
**mir-17**	DNA repair	STAT3- MGMT	upregulated	[87]
**mir-370-3p**	DNA repair	MGMT	upregulated	[88]
**mir-7**	EGFR	EGFR	downregulated	[69]
**mir-7**	PTEN-PI3K-Akt	P13K- AKT	downregulated	[69]
**mir-26**	PTEN-PI3K-Akt	PTEN	up-regulated	[34]
**mir-34a**	EGFR	EGFR	downregulated	[55]
**mir-200a-3p**	PTEN-PI3K-Akt	Gαi1-AKT	downregulated	[72]

## Data Availability

Not applicable.

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
