# Peer review of "MicroRNAs Regulate Cell Cycle and Cell Death Pathways in Glioblastoma"

_ijms, 2021, doi:10.3390/ijms222413550_

Round 1

Reviewer 1 Report

Comments are attached

Author Response

Responses are attached 

Reviewer 2 Report

In this review, the authors summarized the functions of miRNAs in GBM pathways linked to dysregulation of cell cycle control, apoptosis and resistance to treatment highliting a possible role of miRNAs as treatment and prediction biomarkers in clinical setting.

The manuscript is well written although there are some typos that should be corrected. Furthermore, the miRNA nomenclature should be standardized in the text. For example, in table 1 miRNA are reported as MiR-137, MiR-124, etc while in the text are reported as mir-137, miR-124, etc. Similarly, in the Figure 1.

Author Response

Responses are attached 
